# Combining Dynamic Hyperinflation with Dead Space Volume during Maximal Exercise in Patients with Chronic Obstructive Pulmonary Disease

**DOI:** 10.3390/jcm9041127

**Published:** 2020-04-15

**Authors:** Ming-Lung Chuang

**Affiliations:** 1Division of Pulmonary Medicine and Department of Internal Medicine, Chung Shan Medical University Hospital, Taichung 40201, Taiwan; yuan1007@ms36.hinet.net; Tel.: +886-4-2473-9595 (ext. 34718); 2School of Medicine, Chung Shan Medical University, Taichung 40201, Taiwan

**Keywords:** incremental exercise test, plethysmography, diffusing capacity, air trapping, tidal volume and total lung capacity ratio, end-expiratory lung volume

## Abstract

Physiological dead space volume (V_D_) and dynamic hyperinflation (DH) are two different types of abnormal pulmonary physiology. Although they both involve lung volume, their combination has never been advocated, and thus their effect and implication are unclear. This study aimed (1) to combine V_D_ and DH, and (2) investigate their relationship and clinical significance during exercise, as well as (3) identify a noninvasive variable to represent the V_D_ fraction of tidal volume (V_D_/V_T_). Forty-six male subjects with chronic obstructive pulmonary disease (COPD) and 34 healthy male subjects matched for age and height were enrolled. Demographic data, lung function, and maximal exercise were investigated. End-expiratory lung volume (EELV) was measured for the control group and estimated for the study group using the formulae reported in our previous study. The V_D_/V_T_ ratio was measured for the study group, and reference values of V_D_/V_T_ were used for the control group. In the COPD group, the DH_peak_/total lung capacity (TLC, DH_peak_%) was 7% and the EELV_peak_% was 70%. After adding the V_Dpeak_% (8%), the V_D_DH_peak_% was 15% and the V_D_EELV_peak_% was 78%. Both were higher than those of the healthy controls. In the COPD group, the V_D_DH_peak_% and V_D_EELV_peak_% were more correlated with dyspnea score and exercise capacity than that of the DH_peak_% and EELV%, and had a similar strength of correlation with minute ventilation. The V_Tpeak_/TLC (V_Tpeak_%), an inverse marker of DH, was inversely correlated with V_D_/V_T_ (R^2^ ≈ 0.50). Therefore, we recommend that V_D_ should be added to DH and EELV, as they are physiologically meaningful and V_Tpeak_% represents not only DH but also dead space ventilation. To obtain V_D_, the V_D_/V_T_ must be measured. Because obtaining V_D_/V_T_ requires invasive arterial blood gases, further studies on noninvasive predicting V_D_/V_T_ is warranted.

## 1. Introduction

In the alveolar dead space (V_D_) of the three component (Riley) model [1], if alveolar V_D_ exists, residual volume is expected to increase, potentially causing air trapping and hyperinflation of the lung. However, the physiological V_D_ refers to ventilation not involved in gas exchange and involved in unperfused or underperfused alveoli [2] and includes anatomical and alveolar V_D_s [1]. Acute dynamic hyperinflation (DH) refers to a temporary increase in operating lung volume above the resting value, i.e., end-expiratory lung volume at peak exercise (EELV_peak_) [3,4,5,6] minus resting EELV (EELV_rest_) [7]. Because the definitions of alveolar V_D_ and DH are different, physiological V_D_ would not cause DH, and thus their relationship is unclear.

The physiological V_D_/tidal volume ratio (V_D_/V_T_) can be calculated using the Bohr-Enghoff equation [2]. Therefore, V_D_ can be considered to be a part of V_T_, and anatomical V_D_ can be assumed to occur at the beginning of V_T_. Accordingly, as EELV is immediately followed by tidal breathing, beginning V_D_ not included in EELV should be added.

In patients with chronic obstructive pulmonary disease (COPD), the V_D_/V_T_ is often highly increased at rest and usually mildly decreased during exercise as compared with normal subjects. This phenomenon has been hypothesized to be due to a small increase in V_D_ and a small expansion in V_T_, as V_T_ is constrained by DH. V_T_ “floats” above DH and is concomitantly limited by the ceiling of total lung capacity (TLC) and causes reductions in inspiratory reserve volume and O’Donnell threshold [8]. This is in contrast to healthy subjects, in whom a small change in V_D_ and a large increase in V_T_ are usually noted. 

Although the definition and mechanism of V_D_ and DH are quite different, both are volumes; DH, i.e., EELV_peak_ minus EELV_rest_ has been reported to be correlated with the V_D_/V_T_ ratio [3,9,10] (see the Appendix A
Table A1), and EELV_peak_ has been shown to be inversely related to V_Tpeak_/TLC (V_Tpeak_%) [11]. Hence, the aims of this study were as follows: (1) to combine V_D_ with DH; (2) to investigate the relationship between DH and V_D_; (3) to investigate the relationship between V_D_DH and dyspnea, exercise capacity, and ventilation capability; and (4) to investigate the relationship between V_D_/V_Tpeak_ and V_Tpeak_% during maximal exercise in order to find a surrogate for V_D_/V_Tpeak_, which is an invasive variable. This study could help clinicians better understand the relative positions of EELV, DH, V_D_, and V_T_ in TLC, and show that V_D_ and DH together are unfavorable lung volumes during exercise [9,10]. Using the easily calculated V_Tpeak_% during exercise, testing could possibly reflect the invasively measured V_D_/V_Tpeak_, and thus clinicians could use the V_Tpeak_% as an indicator of DH and also V_D_/V_Tpeak_. To the best of our knowledge, this is the first study to integrate the concept of dead space ventilation and DH during exercise.

## 2. Methods

### 2.1. Study Design

In this observational cross-sectional study, we analyzed lung function data and cardiopulmonary exercise with inspiratory capacity maneuver data from subjects with COPD and healthy controls at the Chung Shan Medical university hospital. The relationships between V_Tpeak_% and V_D_/V_T_ were investigated in the subjects with COPD. V_D_, V_T_, and EELV as % of TLC were illustrated using percentages. Signed informed consent was obtained from each participant. The local Institutional Review Board of the institution (CS16174) approved this study, which was conducted in compliance with the Declaration of Helsinki.

### 2.2. Subjects

Subjects aged ≥40 years without any chronic diseases including uncontrolled diabetes mellitus, uncontrolled hypertension, anemia (hemoglobin <13 g/dL), and no acute illnesses in the recent period of 1 month were enrolled. Anthropometric measurements, leisure/sports activities, and cigarette smoking were recorded. Subjects with a body mass index ≤18 kg/m^2^ or ≥32 kg/m^2^ or with laboratory findings of cardiovascular, hematological, metabolic, or neuromuscular diseases were excluded. All of the participants performed lung function and cardiopulmonary exercise tests (CPET). Subjects who did not have sufficient motivation to perform CPET were also excluded.

#### 2.2.1. Study Group

Male adult subjects who underwent spirometry, plethysmography, and diffusing capacity were enrolled if their forced expired volume in one second (FEV_1_)/forced expired capacity (FVC) was <0.7 [12]. The diagnosis of COPD was made according to the global initiative for chronic obstructive lung disease (GOLD) criteria [12]. As few female subjects met the criteria of COPD, they were not included in this study.

#### 2.2.2. Control Group

A group of healthy subjects was recruited among the hospital staff and from the local community through personal contacts. Healthy male subjects reported no chronic diseases.

### 2.3. Measurements

#### 2.3.1. Functional Daily Activity

The oxygen cost diagram (OCD) was used to evaluate the participants’ functional activity. The participants were asked to indicate a point on an OCD, a 100 mm long vertical line with everyday activities listed alongside the line, above which breathlessness limited them [13]. The distance from zero was measured and scored.

#### 2.3.2. Pulmonary Function Testing

Cigarette smoking, drinking coffee, tea, or alcohol, and taking medications were not permitted 24 h before any test. Bronchodilators were not administered within 3 h for short-acting beta agonists and 12 h for long-acting beta agonists before the tests [14,15]. FEV_1_, TLC, residual volume (RV), and diffusing capacity for carbon monoxide (D_L_CO) were measured using spirometry, body plethysmography, and the single-breath technique, respectively, in accordance with the currently recommended standards [16,17,18]. All of the spirometry data were obtained before and after inhaling a standard dose of fenoterol HCl. Post-dose measurements were performed 15 min after inhalation. Static lung volume data and D_L_CO data were obtained before inhaling fenoterol. Simple volume calibration was conducted and accuracy checks for body plethysmograph mouth flow and pressure and box pressure were performed as reported previously [14,15].

#### 2.3.3. Cardiopulmonary Exercise Testing (CPET)

Each subject completed an incremental exercise test using a cycle ergometer to the limit of the symptom. Work rate was selected at a rate of 5–20 W/min based on a derived protocol formula according to the oxygen-cost diagram scores [19]. Oxygen uptake (VO_2_) (mL/min), CO_2_ output (VCO_2_) (mL/min), and minute ventilation (V_E_) were continuously measured. V_O2peak_ was symptom-limited peak V_O2_, because V_O2max_, which was the plateau of V_O2_, was likely not attained in the participants with COPD. The ratio of compartment of TLC and TLC was remarked as the % of TLC such as EELV%, DH%, V_D_%, and V_T_%. A dyspnea score was obtained using the Borg scale by asking the patients about their dyspnea levels while they were performing the ramp-pattern exercise at the end of each minute and at peak exercise.

#### 2.3.4. Dynamic Inspiratory Capacity (IC) Measurement

The techniques used for performing and accepting IC measurements of our previous study [11] were modified from a previous report [7]. Dynamic IC was measured at the end of a steady-state resting baseline, near the middle of loaded exercise (supposed to be near anaerobic threshold, AT), and near peak exercise. Dynamic IC near AT was measured approximately 5–6 min after the start of loaded exercise. EELV was calculated as TLC minus dynamic IC [5,6,20,21]. DH referred to end-expiratory lung volume at AT or peak exercise (EELV_AT or peak_) minus resting EELV (EELV_rest_). In this study, EELV was estimated for subjects with COPD using the formulae from the data of our previous report [11]. EELV_rest_% = 0.7235 − 1.0053 × V_Trest_%; EELV_AT_% = 0.9877 − 2.0132 × V_T AT_%; EELV_peak_% = 0.9491 − 1.35178 × V_Tpeak_%; O’Donnell threshold (OT) = TLC – EELV − V_Tpeak_ (see O’Donnell threshold in Reference [22]).

#### 2.3.5. V_D_/V_T_ Calculation

Brachial artery blood samples were drawn via an arterial catheter connected to a pressure transducer within the last 15 s of each minute after the start of exercise to the peak of exercise [23]. At rest, near the anaerobic threshold, and at the peak of exercise, the physiological V_D_/V_T_ was calculated using a standard formula as follows [24]: V_D_/V_T_ = (P_a_CO_2_ − P_Ē_CO_2_)/P_a_CO_2_ − V_Dm_/(V_T_ − V_Dm_), where P_Ē_CO_2_ = VCO_2_/V_E_ × (P_B_ − 47 mmHg) and PB is barometric pressure measured daily and V_Dm_ is breathing valve dead space. Hemoglobin and biochemistry data were provided. In normal subjects, mean values of V_D_/V_T_ are 0.30 ± 0.08 at rest, 0.20 ± 0.07 at AT, and 0.19 ± 0.07 at peak [2].

### 2.4. Statistical Analysis

Data were summarized as mean ± standard deviation. The sample size was estimated to be at least 17 for each group when the population mean difference in V_D_/V_T_ was 0.1 with a standard deviation for the normal and COPD groups of 0.1 and with a significance level of 0.05 and a power of 0.8. The unpaired t-test was used to compare the means between two groups. The paired t-test was used to compare two related means between two different time points with Bonferroni correction. Pearson’s correlation coefficients were further used when appropriate for quantifying the pairwise relationships among the interested variables. All statistical analyses were performed using SAS statistical software 9.4 (SAS Institute Inc., Cary, NC, USA). Statistical significance was set at *p* < 0.05 and *p* < 0.017 for Bonferroni correction.

## 3. Results

A total of 81 male subjects were enrolled, including 46 subjects (mean age 65.2 ± 5.8 years) with COPD after excluding one subject due to poor motivation, and 34 healthy subjects matched for age and height (mean age 62.2 ± 9.2 years) (Table 1 and Figure 1). Most of the COPD subjects had GOLD stages II and III with hyperinflation and air trapping, normocapnia, and borderline hypoxemia at rest and could perform daily brisk walking on the level. Compared to the healthy controls during exercise, most of the COPD subjects had mildly impaired exercise capacity due to ventilatory limitation with poor lung expansion, significant oxyhemoglobin desaturation, and exercise hyperventilation (Table 2).

### 3.1. The % of TLC: EELV%, DH%, V_D_%, V_T_%, V_D_DH%, V_D_EELV%, and V_T_EELV% (or End-Inspiratory Lung Volume, EILV)

In the COPD group, EELV_rest_% was 63% ± 2% and EELV_peak_ was 70% ± 7% as compared with 48% ± 13% and 46% ± 13% in the healthy group (Figure 2, group comparisons, both *p* < 0.0001). Hence, DH_peak_% was 7% ± 7% as compared with 1% ± 10% in the healthy group (*p* = 0.03). In the COPD group, V_Drest_% was 5% ± 1% and V_Dpeak_% was 8% ± 2% as compared with 4% ± 2% and 6% ± 1% in the healthy group (Figure 2, group comparisons: *p* < 0.01 and *p* < 0.0001). In the COPD group, DH_peak_% was similar to V_Dpeak_% at peak exercise (7% ± 7% vs. 8% ± 2%, *p* = 0.61).

After combining V_D_ with DH (V_D_DH%), V_D_DH_rest_% was 5% ± 1% and V_D_DH_peak_% was 15% ± 5% in the COPD group as compared with 4% ± 2% and 7% ± 10% in the healthy group (group comparisons, both *p* < 0.01). After combining V_D_ with EELV (V_D_EELV%), V_D_EELV_rest_% was 68% ± 1% and V_D_EELV_peak_% was 78% ± 6% in the COPD group as compared with 52% ± 13% and 52% ± 13% in the healthy group (group comparisons, both *p* < 0.0001). After combining V_T_ with EELV (V_T_EELV% or EILV%), V_T_EELV_rest_% was 72% ± 0% and V_T_EELV_peak_% was 88% ± 2% in the COPD group as compared with 62% ± 13% and 78% ± 14% in the healthy group (group comparisons, *p* < 0.01 and *p* < 0.001, respectively).

### 3.2. Relationships among the Compartments of TLC

V_Dpeak_% was moderately positively correlated with V_Tpeak_% (Table 3, r = 0.66, *p* <0.0001) and moderately negatively correlated with the other compartments at peak exercise (r = −0.47 to −0.68, *p* <0.01 to <0.0001).

### 3.3. Relationships between the % of TLC and Oxygen Uptake, Minute Ventilation, and Dyspnea

In the % of TLC, V_D_EELV_peak_% and V_D_DH_peak_% showed the best correlations with ΔBorg/ΔVCO_2_ and, and a similar strength of correlation with V_Epeak_ (Table 3). The higher the V_D_DH_peak_% and V_D_EELV_peak_%, the higher the dyspnea score and the lower the VO_2peak_% and V_Epeak_.

### 3.4. V_Tpeak_% versus V_D_/V_Tpeak_

In the COPD group, V_Trest_% was 9% ± 2% and V_Tpeak_% was 18% ± 5% as compared with 13% ± 7% and 32% ± 54% in the healthy group (Figure 2, group comparisons *p* < 0.01 and *p* < 0.0001). In the COPD group, there was a negatively significant relationship between V_T_% and V_D_/V_T_ at rest, anaerobic threshold, and peak exercise, and this was stronger as the exercise intensity increased (see the Appendix A
Table A2, r = −0.34 to −0.64, *p* = 0.02 to *p* < 0.0001). When pooling the data of these two variables at the three time points together, the relationship was much closer (r = −0.72, *p* < 0.0001).

## 4. Discussion

There are four main findings in this study. First, V_D_ and DH (V_D_DH) and V_D_ and EELV (V_D_EELV) could be combined. Secondly, we found that in the patients with COPD, V_D_ and DH were similar in size, and that V_D_EELV_rest_ accounted for 68% of the TLC and V_D_EELV_peak_ accounted for up to 78%. Third, compared to DH_peak_% and EELV_peak_%, V_D_DH_peak_% and V_D_EELV_peak_% were more closely related to dyspnea and exercise capacity and had a similar power in relation to ventilation capability. Lastly, V_Tpeak_%, a recently reported marker of DH_peak_ [11], was moderately negatively correlated with V_D_/V_Tpeak_. To the best of our knowledge, these findings have not previously been published.

### 4.1. The % of TLC

The importance of EELV_peak_% has been reported when the EELV_peak_ is ≥75% of TLC, a threshold value which can maximize the sensitivity and specificity of detecting ≤5.5 mL/heartbeat change in oxygen pulse (ΔO_2P_) and ≤10,000 oxygen uptake efficiency slope (OUES) during exercise [25], where ΔO_2P_ and OUES are markers of cardiovascular function. In addition to EELV_peak_% >75% [25], the reciprocal IC_peak_/TLC <25% [26] has also been associated with lower O_2P_ and exercise capacity in patients with severe COPD. IC_peak_/TLC <23% has also been associated with lower O_2P_ and exercise capacity in patients with severe COPD [27]. Although OUES was not measured in this study, our previous study reported that IC_peak_/TLC was significantly correlated with O_2P_ and ΔO_2P_ (r = 0.35–0.36, both *p* < 0.05) [28]. These results support an interaction between hyperinflation and decreased cardiac function that can contribute to exercise limitation in these patients. A greater amount of trapped gas in the lung increases the intrinsic positive end-expiratory pressure, and this compresses the heart and impedes venous return causing further heart impairment [25,26]. It has recently been reported that this compression can occur even at rest [29].

DH has been shown to increase with exercise in patients with COPD [3,4,5,6,9,10,20,21,22], and thus EELV caused failure of V_T_ to expand, as in the healthy subjects in this study (0.6 ± 0.31 L versus 1.12 ± 0.57 L, *p* < 0.0001). A high level of V_D_EELV “buoyed” the expandable basic lung volume above its position, meaning that V_T_ had limited room to expand downwards so that it could not help but invade upwards to the OT or near its limit (Figure 2). In COPD, decreased OT [3,22] and increased DH have been reported to be possible causes of exercise limitation [30], although some studies have questioned whether DH occurs in all COPD patients [31,32,33]. These previous studies have measured DH_peak_ but not included V_Dpeak_. In this study, V_D_DH_peak_% and V_D_EELV_peak_% were slightly better than DH_peak_% and EELV_peak_% with regards to the correlation with ΔBorg/ΔVO_2_ and VO_2peak_% and had a similar power with regards to the correlation with V_Epeak_ (Table 3). Therefore, it could be reasonable to combine V_Dpeak_ with DH_peak_ and to combine V_Dpeak_ with EELV_peak_. In this study, V_D_EELV_peak_%, an unfavorable lung volume, was elevated to as high as 78% ± 6% of TLC.

In the patients with COPD in this study, although V_Dpeak_% was small as compared with EELV_peak_% but similar to DH _peak_% in size, V_D_DH _peak_% accounted for 15% of TLC. The majority of the increase in physiological V_D_ must have come from alveolar V_D_, as the increase in anatomical V_D_ was estimated to be only 12 mL and 20 mL in the COPD and control groups, respectively, based on the estimation that anatomical V_D_ would increase 20 mL per liter increase in EELV [1]. Hence, the remaining increase in physiological V_D_ must have come from alveolar V_D_, which is strongly influenced by lung pathology but less influenced by other factors such as age, sex, body size (1 mL of physiological dead space per pound of weight reported by Radford), posture, low cardiac output, pulmonary emboli, and posture [1].

V_D_% and EELV% were moderately negatively correlated (Table 3). This is because V_D_% and V_T_% were moderately positively correlated and V_T_% and EELV% were highly negatively correlated (r = −0.83, *p* < 0.0001) [11]. V_D_% was positively correlated with V_T_% because V_D_ is calculated by V_D_/V_T_ multiplied by V_T_. Hence, the larger the V_T_, the larger the V_D_, and the smaller the EELV. It is clear that V_D_ is different from EELV and DH in the direction of correlation, that these volumes can be combined, and that the combinations are more related to exercise capacity and exertional dyspnea sensation, although V_D_ is small. Interestingly, V_D_% alone was poorly related to exercise tolerance and dyspnea. However, the relationships between DH% and EELV% versus exercise tolerance and dyspnea were slightly improved after adding V_D_% (Table 3).

### 4.2. V_T_% versus V_D_/V_T_

V_D_/V_T_ has been reported to be the most consistent gas exchange abnormality in smokers with only mild abnormalities in spirometry [3]. However, invasive methods to obtain arterial blood gases are needed to measure V_D_/V_T_. In this study, V_T_%, an inverse marker of DH [11], was inversely correlated with V_D_/V_T_ (R^2^ ≈ 0.50) (see the Appendix A
Table A2). However, Mahut et al. reported that V_D_/V_Tpeak_ was only mildly correlated to DH (r = −0.45, *p* = 0.004) [10], where DH was represented by IC_peak_% predicted [10]. This difference in correlation between DH and V_D_/V_T_ in these two studies could be due to the different criteria used for DH, i.e., IC_peak_% predicted versus V_T_%. Predicted IC data were obtained from the general population, whereas V_T_% was directly measured in the participants. In addition, Mahut et al. reported that the alveolar volume (V_A_)/TLC ratio was significantly correlated with V_D_/V_Trest_ but much less significantly correlated with V_D_/V_Tpeak_ (see the Appendix A
Table A1) [10]. V_A_ is usually measured using the single breath helium dilution method at rest and is equal to TLC − V_D_ [34]. Therefore, V_A_ would underestimate TLC in subjects with poorly communicating airways or disequilibrium of ventilation. V_A_/TLC measured at rest cannot reflect DH_peak_, so that it was poorly correlated with V_D_/V_Tpeak_. Moreover, in this study, the relationship between V_T_% and V_D_/V_T_ was strongest when data at rest, anaerobic threshold, and peak exercise were pooled (see the Appendix A
Table A2, r = −0.72, *p* < 0.0001). The mechanism underpinning the stronger relationship between V_Tpeak_% and V_D_/V_Tpeak_ with increasing exercise intensity could be due to the common factor V_Tpeak_ being highly constrained at peak exercise. The stronger relationship between V_T_% and V_D_/V_T_ after pooling different stages of exercise is comparable to a previous study in which V_E_/VCO_2_ was used instead of V_T_% in healthy subjects and patients with COPD [3].

Nevertheless, Paoletti et al. reported that V_Tpeak_/FEV_1_ > 1 (or V_Tpeak_/IC = 0.96 ± 0.05), emphysema, the slope of V_E_/VCO_2_, and P_ET_CO_2peak_ values were colinear [35] (Figure 3). In their study, the patients with COPD had high RV% predicted and high emphysema score measured with high resolution computed tomography (HRCT). They hypothesized that V_Tpeak_/FEV_1_ > 1 or elevated V_Tpeak_/IC was due to DH occurring at peak exercise in patients with severe emphysema, which is comparable with our study and another study using V_Tpeak_/SVC to assess the severity of emphysema evaluated with HRCT [36] (Figure 3). However, it has been reported that the change in V_D_/V_T_ from rest to peak exercise was not related to the severity of emphysema [35]. In the current study, V_Tpeak_/FEV_1_ > 1 and V_Tpeak_/SVC were correlated with V_Tpeak_%, respectively (Figure 3, r = −0.36 and 0.66, *p* = 0.001, *p* < 0.0001), however neither were correlated with V_D_/V_Tpeak_. Nevertheless, V_Tpeak_% was correlated with V_D_/V_Tpeak_ (r = −0.64, *p* < 0.0001), suggesting that V_Tpeak_% could be more powerful than V_Tpeak_/FEV_1_ and V_Tpeak_/SVC (Figure 3).

### 4.3. Clinical Implications of V_D_DH_peak_% and V_D_EELV_peak_%, and V_Tpeak_%

Since DH may not occur in all COPD patients [31,32,33], as V_D_DH_peak_% and V_D_EELV_peak_% are substantially larger and slightly more related to dyspnea [31] and exercise capacity than DH% and EELV%, and as V_Tpeak_% can be obtained easily and noninvasively, these three markers could potentially be used to evaluate the effect of bronchodilator or lung volume reduction surgery on dyspnea and exercise tolerance.

## 5. Study Limitations

Airflow obstruction should be defined as a FEV_1_/VC ratio below the fifth percentile (z-score −1.645) of the distribution of a reference population [17] according to the 2019 ATS-ERS technical statement [16]. In the present study, the use of GOLD criteria to define COPD could have introduced age, sex, and height selection bias. However, the severity of most of the subjects with COPD in this study had GOLD stages II–IV (93.5%), and thus the likelihood of underdiagnosing COPD was small. Although OCD is not a commonly used tool to evaluate physical activity for patients with COPD, previous studies have suggested that the OCD and the COPD assessment test should be used simultaneously when undertaking clinical evaluations of patients with COPD, and that the OCD in ramp-slope selection should be used for dyspneic patients when undertaking CPET [13,19]. However, the International Physical Activity Questionnaire and accelerometry could also be helpful in this case [37,38]. A novel analytical method reported calculating shunt V_D_ by subtracting respiratory V_D_ (i.e., anatomical V_D_ and alveolar V_D_) from physiological V_D_ [39]. We did not calculate shunt V_D_, as this method is sophisticated and the shunt V_D_ level was expected to be small. Tidal flow limitation measured with negative expiratory pressure has been shown to play a role in reducing the IC at rest, during which tidal flow limitation constrains V_T_ expansion during exercise thereby causing an elevation in V_D_/V_T_ at peak exercise [40]. Although tidal flow limitation was not measured in this study, it can be anticipated to occur in the subjects with more severe airflow obstruction and higher air trapping with a lower IC [41]. In the COPD group in this study, EELV was estimated using the formulae reported in our previous study [11], and thus the estimated DH% and EELV% values may not be exactly the same as the measured data. In the healthy controls, data on V_D_/V_T_ at rest, AT, and peak exercise were retrieved from reference subjects, as it was difficult to obtain permission from our Institutional Review Boards to perform arterial catheterization for exercise testing. The emphysematous phenotype could be related to V_D_DH. However, as there were relatively few subjects and emphysema was not evaluated using HRCT in this study, further studies are warranted to address these issues. Lastly, V_D_ cannot be obtained without using invasive method in patients with COPD, and thus its clinical implication could be limited. Studies to investigate the development of a novel noninvasive method to obtain V_D_ or V_D_/V_T_ are warranted. Finally, using Jones’ and Bohr’s equations to estimate V_D_/V_T_ in subjects with COPD is not suitable, as P_ET_CO_2_ used in the equations cannot be used as a surrogate for P_a_CO_2_ or alveolar PCO_2_ [42,43].

## 6. Conclusions

Although the definitions of V_D_ and DH are quite different, this study shows the utility of their combination, and that it could play a role in physiology with regards to the evaluation of exertional dyspnea and exercise capacity in subjects with COPD. In addition, V_T_% was significantly correlated with V_D_/V_T_, suggesting that V_T_% is not only a convenient marker for DH as reported previously, but also a potential noninvasive marker for V_D_/V_T_.

## Figures and Tables

**Figure 1 jcm-09-01127-f001:**
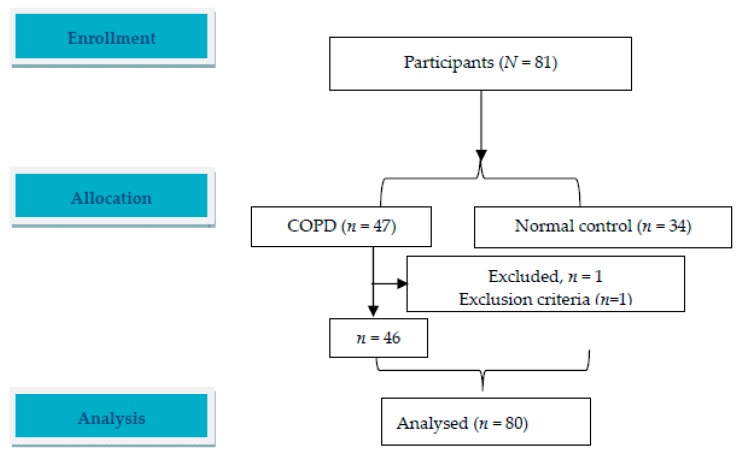
Flow diagram. A total of 81 participants with chronic obstructive pulmonary disease and healthy controls were screened.

**Figure 2 jcm-09-01127-f002:**
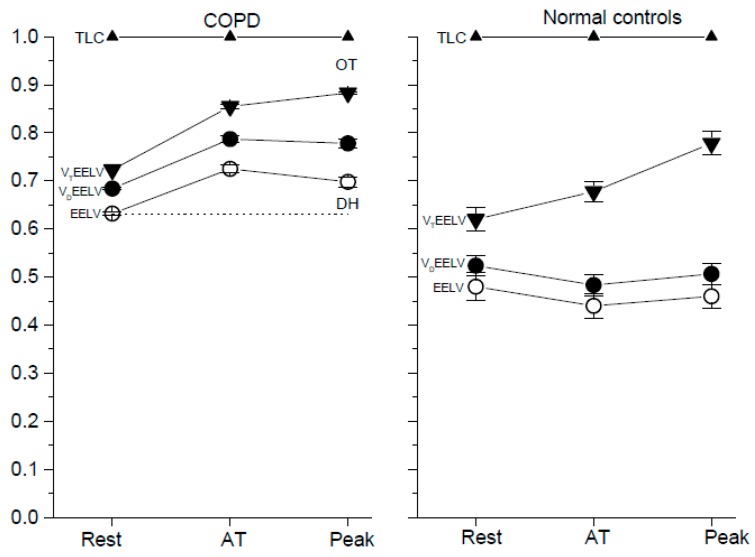
The % of total lung capacity (TLC, upward triangles) at rest, anaerobic threshold (AT) and peak exercise. Left panel COPD group and right panel normal controls. Open circles, end-expiratory lung volume (EELV); solid circles, dead space volume (V_D_) plus EELV; down triangles, tidal volume (V_T_) plus EELV (i.e., end-inspiratory lung volume, EILV); vertical bars, standard error of estimate; OT, O’Donnell threshold; DH, dynamic hyperinflation indicating EELV at AT or peak exercise minus EELV at rest; dashed line, EELV at rest. Comparisons of each compartment between COPD patients and normal controls at rest, AT and peak exercise, respectively, all *p* < 0.0001 except V_T_EELV at rest, *p* < 0.01 and V_T_EELV at peak exercise, *p* < 0.001. In COPD patients, comparisons of each compartments of TLC between two time points, all *p* < 0.0001 except EELV at AT versus EELV at peak exercise, *p* < 0.001 and V_D_EELV at AT versus V_D_EELV at peak exercise, *p* = 0.046, which was insignificant.

**Figure 3 jcm-09-01127-f003:**
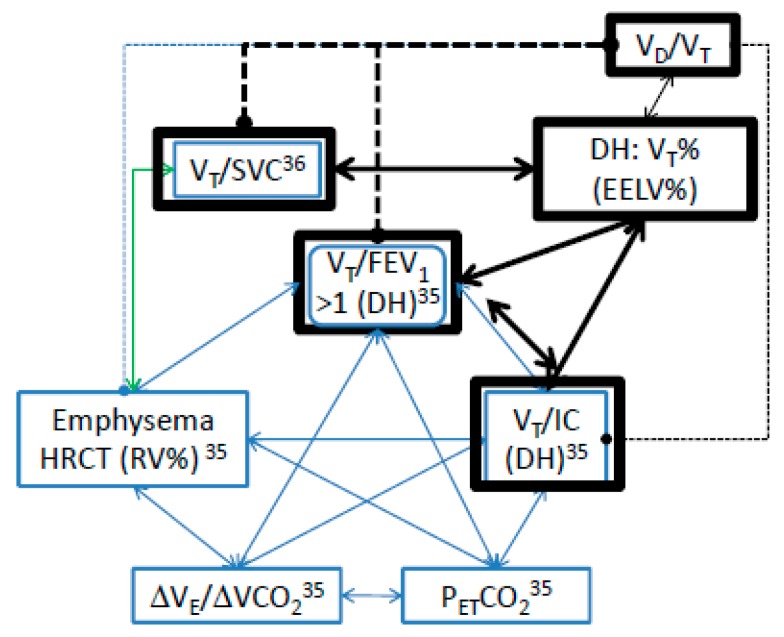
Relationships among dynamic hyperinflation (DH) variables and relationships between DH variables and dead space fraction (V_D_/V_T_) in patients with chronic obstructive pulmonary disease. Black bolded boxes, from this study; blue boxes, from References [35,36]. Solid lines, significantly correlated; dashed lines, not significantly correlated. Black lines, from this study; blue lines, from reference [35]; green line, from reference [36]. V_T_%, tidal volume and total lung capacity (TLC) ratio; EELV%, end-expiratory lung volume and TLC ratio; V_T_/SVC, V_T_ and slow vital capacity ratio; V_T_/FEV_1_, V_T_ and forced expired volume in one second ratio; HRCT, high resolution computed tomography; RV%, residual volume predicted %; Δ V_E_/Δ VCO_2_, slope of minute ventilation and CO_2_ output; P_ET_CO_2_, end-tidal CO_2_ pressure.

**Table 1 jcm-09-01127-t001:** Demographics and lung function in 80 male subjects with 46 subjects of chronic obstructive pulmonary disease (COPD) and 34 healthy subjects.

	COPD		Normal Controls		
	Mean	SD	Mean	SD	*p*
Age, years	65.2	5.8	62.2	9.2	0.10
Height, cm	165.0	6.4	167.0	5.3	0.14
Weight, kg	60.4	11.2	69.2	8.9	0.0002
Body mass index, kg/m^2^	22.1	3.5	24.8	2.7	0.0003
Cigarette smoke, pack⋅year	42.3	19.2	4.7	17.4	<0.0001
Oxygen cost diagram, cm	7.0	1.4	8.3	1.0	<0.0001
TLC% predicted, %	135	21	97	11	<0.0001
RV% predicted, %	200	55	101	17	<0.0001
RV/TLC	0.58	0.09	0.39	0.06	<0.0001
IC% predicted, %	92	27	99	17	0.15
D_L_CO% predicted, %	69	22	106	16	<0.0001
FVC% predicted, %	81	21	101	13	<0.0001
FEV_1_% predicted, %	50	19	103	13	<0.0001
GOLD, I, II, III, IV, n	3, 18, 19, 6		NA		NA
FEV_1_/FVC	0.49	0.13	0.93	0.28	<0.0001
Hemoglobin, g/dL	14.8	1.5	14.6	1.2	0.78
Creatinine, mg/dL	1.1	0.2	1.0	0.3	0.25
Na^+^, meq/L	140.5	2.4	138.4	2.2	0.73
K^+^, meq/L	4.3	0.5	4.1	0.4	0.52
Albumin, mg/dL	4.2	0.4	NA	NA	NA
pH	7.40	0.03	NA	NA	NA
P_a_CO_2_, mmHg	40.6	6.4	NA	NA	NA
P_a_O_2_, mmHg	79.3	10.1	NA	NA	NA
S_P_O_2_, %	95.3	2.6	97.2	1.2	<0.0001

TLC: total lung capacity, RV: residual volume, IC: inspiratory capacity, D_L_CO: diffusing capacity for carbon monoxide, FVC: forced vital capacity, FEV_1_: forced expired volume in one second., GOLD: global initiative for chronic obstructive lung disease, S_P_O_2_: oxyhemoglobin saturation measured with pulse oximetry. NA: not available or not applicable.

**Table 2 jcm-09-01127-t002:** Cardiopulmonary exercise test at peak exercise in male subjects with chronic obstructive pulmonary disease (COPD) (*n* = 46) and male healthy subjects (*n* = 34).

	COPD		Normal Controls		*p*
	Mean	SD	Mean	SD	
Work rate, watts	91.8	42.9	146.6	34.7	<0.0001
% predicted	69	30	115.9	22.9	<0.0001
Oxygen uptake (VO_2_), mL/min	1073	355	1708	402	<0.0001
% predicted	69.3	20.9	90.7	19.4	<0.0001
Anaerobic threshold, mL/min	489	137	1018	302	<0.0001
%V_O2max_ predicted, %	31.1	8.0	53.0	11.8	<0.0001
Respiratory exchange ratio	1.05	0.10	1.16	0.14	0.0003
Cardiac frequency, b/min	133	20	149	17	0.0002
% predicted max, %	81.3	12.0	94.7	9.6	<0.0001
Oxygen pulse, mL/min	8.1	2.4	11.5	2.5	<0.0001
% predicted	85.3	23.5	96.7	22.9	0.03
Minute ventilation V_E_/V_O2nadir_	36.9	8.0	28.2	3.9	<0.0001
S_P_O_2_,%	91.0	5.8	96.8	1.2	<0.0001
V_E_, L/min	38.6	12.3	70.4	18.0	<0.0001
V_E_/MVV	1.16	0.36	0.63	0.15	<0.0001
Breathing frequency, breath/min	32.6	5.9	36.6	9.3	0.03
Tidal volume (V_T_), L	1.19	0.35	1.96	0.42	<0.0001
V_T_/total lung capacity (TLC)	0.19	0.05	0.32	0.05	<0.0001
Dead space volume (V_D_)/V_T_	0.43	0.10	0.19 *	0.07	NA
pH	7.32	0.04	NA		NA
P_a_CO_2_, mmHg	46.1	7.8	NA		NA
P_a_O_2_, mmHg	71.0	16.7	NA		NA

Oxygen pulse = VO_2_/cardiac frequency; oxyhemoglobin saturation measured with pulse oximetry—S_P_O_2_; maximum voluntary ventilation—MVV; * from Reference [2]. NA: not applicable or not available.

**Table 3 jcm-09-01127-t003:** Relationships among the compartments of total lung capacity (TLC) and correlations of seven components of total lung capacity (TLC) with oxygen uptake (VO_2_), minute ventilation (V_E_), and dyspnea at peak exercise in 46 patients with COPD.

Peak	V_D_%	VO_2_	V_E_	ΔBorg/ΔVO_2_
EELV%	−0.67 ^†^	−0.62 ^†^	−0.75 ^†^	0.66 ^†^
DH%	−0.61 ^†^	−0.69 ^†^	−0.78 ^†^	0.72 ^†^
V_D_%	1	0.26 *	0.46 **	−0.19
V_T_%	0.66 ^†^	0.62 ^†^	0.76 ^†^	−0.67 ^†^
V_D_DH%	−0.68 ^†^	−0.74 ^†^	−0.74 ^†^	0.78 ^†^
V_D_EELV%	−0.47 **	−0.74 ^†^	−0.74 ^†^	0.78 ^†^
V_T_EELV%	−0.68 ^†^	−0.60 ^†^	−0.71 ^†^	0.63 ^†^

%: variable divided by TLC, EELV: end-expiratory lung volume, DH: dynamic hyperinflation indicating EELV at peak exercise subtracting resting EELV, V_D_DH: combing dead space (V_D_) and DH, V_T_: tidal volume, Δ: change. * 0.05 > *p* ≤ 0.1, ** *p* ≤ 0.01, ^†^
*p* ≤ 0.0001.

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
