# Peer review of "Combining Dynamic Hyperinflation with Dead Space Volume during Maximal Exercise in Patients with Chronic Obstructive Pulmonary Disease"

_jcm, 2020, doi:10.3390/jcm9041127_

Round 1
Reviewer 1 Report
This cohort study examines a way of measuring physiological gas trapping on exercise in COPD. The background literature including landmark paper on the topic by O-Donnell et al is clearly described in the introduction. Ethical approval was given, consistent with GCP, however I am not clear if the subjects gave individual consent - I expect they would have done as some of the tests clearly go beyond standard care. This should be clarified.
The methods are written clearly and appear appropriate to the research question, with the number of recruited subjects being reasonable (in fact good) for a study of this type. The authors should be congratulated on managing to obtain so much data on this number of COPD patients, especially when exercising. The results are generally well presented and easy to understand. Overall the quality of English is good, and formatting is professional. However there are some points the authors could consider to improve their manuscript.
- Clarify consent
- Oxygen cost diagram - this is not a commonly used tool. Why was it chosen and what advantages/disadvantages might it have over other physical activity tools? (eg IPAQ questionnaire, accelerometry). This could be covered in the study limitations
- Discussion - this is very long. I would favour reducing it in length and putting the clinical messages first, since these are the most likely to be of interest to the readership
Author Response
This cohort study examines a way of measuring physiological gas trapping on exercise in COPD. The background literature including landmark paper on the topic by O-Donnell et al is clearly described in the introduction. Ethical approval was given, consistent with GCP, however I am not clear if the subjects gave individual consent - I expect they would have done as some of the tests clearly go beyond standard care. This should be clarified.
|
2.1. Changed as recommended. |
The methods are written clearly and appear appropriate to the research question, with the number of recruited subjects being reasonable (in fact good) for a study of this type. The authors should be congratulated on managing to obtain so much data on this number of COPD patients, especially when exercising. The results are generally well presented and easy to understand. Overall the quality of English is good, and formatting is professional. However there are some points the authors could consider to improve their manuscript.
- Clarify consent
|
2.1. individual consent: Changed as recommended. |
- Oxygen cost diagram - this is not a commonly used tool. Why was it chosen and what advantages/disadvantages might it have over other physical activity tools? (eg IPAQ questionnaire, accelerometry). This could be covered in the study limitations
|
5. Do as recommended. |
- Discussion - this is very long. I would favour reducing it in length and putting the clinical messages first, since these are the most likely to be of interest to the readership
|
4.2. Some sentences were deleted and edited. |
Reviewer 2 Report
Dr Chuang have studied 81 male subjects with or without COPD. Various lung function, volumes and breathing gases were tested at baseline and during an incremental exercise test. Dead space fraction was also assessed by measuring PaCO2 in the blood collected at various intervals during exercise. Dead space (VD) was added to dynamic hyperinflation (DH) and end-expiratory lung volume (EELV) at three timepoints (baseline, at near aerobic threshold and maximal capacity). Dr Chuang claims that these combined volumes better correlate with other physiological readouts. However, the differences in predictability with the other single volumes that are conventionally used are not very convincing. Dr Chuang also suggests that a less invasive measure correlating with VD+DH or VD+EELV, namely tidal volume in percentage of TLC (Vt%), may be an alternative.
Dr Chuang needs to clearly distinguish between physiological, anatomical and alveolar VDs throughout the manuscript. This should also be clear from the get go (so even in the abstract). As it is, I am confused and my misunderstanding may have been the source of my first major concern.
Majors:
VD in equation 5 includes anatomical VD? One problem is that anatomical VD might or might not expand during a tidal breath and DH depending of airway stiffness. Is it possible to distinguish these changes in anatomical dead space from the alveolar VD?
There are non-invasive ways to estimate anatomical dead space. Perhaps this would help interpreting the results.
Line 122. I don’t understand. Have you used TLC minus IC or the equations that follow?
Minors:
What was the CPET? Cycling?
Line 100. What is the rule?
Line 113. ‘with COPD’, not ‘of COPD’
Instead of stating ‘compartments of TLC’, why not just mentioning at an early point in the manuscript that all the measured volumes are expressed in % of TLC?
Why are you stating ‘standard deviation’ at line 133 and ‘standard error of estimate’ at line 166?
T-tests should not be used to compare between 2 time points within a group when more than two time points were measured.
Line 211. What was the oxygen uptake efficiency slope in your study?
Table 3. It begs the question as to whether a correlation of -.74 is truly better than a correlation of -.69 when they both have a p value of under .0001.
I’m not sure how useful is Figure 3? It is not discussed.
Author Response
Dr Chuang needs to clearly distinguish between physiological, anatomical and alveolar VDs throughout the manuscript. This should also be clear from the get go (so even in the abstract). As it is, I am confused and my misunderstanding may have been the source of my first major concern.
|
Changed as recommended from abstract to the end of the manuscript. |
Majors:
VD in equation 5 includes anatomical VD? One problem is that anatomical VD might or might not expand during a tidal breath and DH depending of airway stiffness. Is it possible to distinguish these changes in anatomical dead space from the alveolar VD?
|
VD herein represents physiological VD. It is not possible to differentiate these changes in anatomical dead space from that in the alveolar VD by Bohr-Enghoff equation. |
There are non-invasive ways to estimate anatomical dead space. Perhaps this would help interpreting the results.
|
Anatomical VD is easily derived by the graphical solution developed by Fowler. However, it would be difficult to measure VO2 when applying 100% FiO2 in order to measure anatomical VD. |
Line 122. I don’t understand. Have you used TLC minus IC or the equations that follow?
|
2.3.4. TLC minus IC was used for normal controls. For COPD, the equations that follow were used. 5. It was reported as the study limitations. |
Minors:
What was the CPET? Cycling?
|
2.3.3. Using a cycle ergometer was added in. |
Line 100. What is the rule?
|
2.3.2. It was changed. Cigarette smoking, drinking coffee, tea, or alcohol, and taking medications were not permitted 24 h before any test. Bronchodilators were not administered within 3 h for short-acting beta agonists and 12 h for long-acting beta agonists before the tests. |
Line 113. ‘with COPD’, not ‘of COPD’
|
2.3.3. Changed as recommended. |
Instead of stating ‘compartments of TLC’, why not just mentioning at an early point in the manuscript that all the measured volumes are expressed in % of TLC?
|
In multiple places, changed as recommended when appropriate. |
Why are you stating ‘standard deviation’ at line 133 and ‘standard error of estimate’ at line 166?
|
2.4. Our statement with SD was used for description purpose of the original data. In Figure 2, the SEE or SE presented in our subsequent analysis was used for quantifying the sampling variation in comparison of population means between groups, and this expression is widely used in graphical display, e.g. We used it in our previous reports, J. Appl. Physiol. 87(3): 1048–1058, 1999, Med. Sci. Sports Exerc., Vol. 34, No. 10, pp. 1614–1623, 2002, and J. Clin. Med. 2019, 8, 1641. |
T-tests should not be used to compare between 2 time points within a group when more than two time points were measured.
|
While we agree that methods such as linear mixed models could handle more complicated situations beyond pairwise comparisons between different time points within the same group of people, we chose to perform paired t test with Bonferroni corrections to control overall type I error. The reasons are (1) The pairwise comparisons were planned a priori, and the overall significance by ANOVA is less interested in our study, (2) Bonferroni correction is a more conservative approach than other methods. We have added the significance level for Bonferroni correction in the Statistical Analysis section. Please see our previous reports, J. Appl. Physiol. 87(3): 1048–1058, 1999 and J. Clin. Med. 2019, 8, 1641. |
Line 211. What was the oxygen uptake efficiency slope (OUES ) in your study?
|
4.1. Although OUES is a marker for cardiovascular function, it was not the key interested variable in this study. ICpeak/TLC was significantly correlated with O2P and DO2P, which are also markers of cardiovascular function. SCIentIFIC REPOrtS | 7: 10929 | DOI:10.1038/s41598-017-11189-x |
Table 3. It begs the question as to whether a correlation of -.74 is truly better than a correlation of -.69 when they both have a p value of under .0001.
|
4.1. L315. It was edited. With relation between two variables, r=-.74 can explain 7% more than r=-0.69, although these two correlations are high. |
I’m not sure how useful is Figure 3? It is not discussed.
|
4.2. L347-358. Figure 3 is discussed herein. |
Reviewer 3 Report
This study aimed (1) to combine VD and DH, and (2) to investigate its clinical significance during exercise and to identify a noninvasive variable to represent VD fraction. Forty-six male COPD subjects and 34 healthy male subjects were enrolled. In the COPD group, VDDHpeak% and VDEELVpeak% were more correlated with dyspnea score and exercise capacity than DHpeak% and EELV% and had a similar strength of correlation with minute ventilation. VT% was inversely correlated with VD/VT. Therefore, we recommend that VD should be added to DH and EELV. VT% was significantly correlated with VD/VT, suggesting that VT% is not only a convenient marker for DH, but also a potential noninvasive marker for VD/VT.
Major Comments
- The authors described “in the COPD group, VDDHpeak% (BS; r=0.78, V(・)O2; r= -0.74) and VDEELVpeak% (BS; r=0.78, V(・)O2; r= -0.74) were more correlated with dyspnea score and exercise capacity than DHpeak% (BS; r=0.72. V(・)O2; -0.69) and EELV% (BS; r=0.66, V(・)O2; r=-0.62)”. However, both the correlation coefficient were very good and there were bit difference. Is there a significant difference in the correlation coefficient between the DH markers and DH+VD markers statistically?
- The degree of DH was different according with phenotype, which is emphysema dominant and non-dominant. The COPD patients with emphysema and hyperinflation show marked DH. Is the association between VDDH and dyspnea or exercise intolerance different between the phenotypes? If the COPD patients without emphysema show different manner in the relationship between VD/VT and DH from the patients with emphysema dominant phenotype, the evaluation of VDDH may be meaningful.
- The measurement of VD/VT is invasively not convenient. However, the measurement of IC as a marker of DH is easy and convenient. If VD/VT is simply and conveniently evaluated by the method of expired gas analysis only, can the same results be obtained?
- The reason why VD/VT has to be measured in addition to the DH is not clear. The worsening of ventilator efficiency may be occurred together with hyperinflation. Is the worsening of VD/VT an independent risk factor for dyspnea and exercise intolerance from DH?
- You mentioned that "although the definitions of VD and DH are quite different, this study shows the utility of their combination, and that it may play a role in physiology with regards to the evaluation of exertional dyspnea and exercise capacity in COPD".I have following comments about it:
- EELV and Borg were positively correlated, while VD and Borg were negatively correlated. Please explain why VDEELV which combined those indexes more correlated to dyspnea, even though those indexes have the different correlation to dyspnea respectively.
- You should also discuss why EELV and VD are negatively correlated. In Figure 2, EELV and DVEELV were expected to be positively correlated because they were almost parallel. Even if the relationship between EELV and VD has not been completely elucidated it in previous study, this is interesting and should be considered.
- These results are not discussed enough even though relevant previous studies are cited in the discussion session.
- You mentioned that VD/VT can be estimated from VT. However, the practical relationship between VT and VD/VT is not discussed such as the reason why VD/VT can be estimated from VT.
Minor comments:
- Please add the method of calculation for sample size
- Table E2 the abbreviations of "EELV" should be deleted because EELV is not included in that table.
- The method of evaluation for important outcomes such as DH and Borg should be explained briefly, not only by citing previous own studies
- There are four main aim of study in "Introduction" session and four main conclusions in "Discussion" session respectively. You should only mention the points what you would like to present.
Author Response
Major Comments
- The authors described “in the COPD group, VDDHpeak% (BS; r=0.78, V(・)O2; r= -0.74) and VDEELVpeak% (BS; r=0.78, V(・)O2; r= -0.74) were more correlated with dyspnea score and exercise capacity than DHpeak% (BS; r=0.72. V(・)O2; -0.69) and EELV% (BS; r=0.66, V(・)O2; r=-0.62)”. However, both the correlation coefficient were very good and there were bit difference. Is there a significant difference in the correlation coefficient between the DH markers and DH+VD markers statistically?
|
4.3. It was edited. With relation between two variables, r=.78 and -.74 can explain 7-9% and 17% more than the others that you mentioned above, although these three correlations are all high. |
- The degree of DH was different according with phenotype, which is emphysema dominant and non-dominant. The COPD patients with emphysema and hyperinflation show marked DH. Is the association between VDDH and dyspnea or exercise intolerance different between the phenotypes? If the COPD patients without emphysema show different manner in the relationship between VD/VT and DH from the patients with emphysema dominant phenotype, the evaluation of VDDH may be meaningful.
|
5. Study limitations. I totally agree with you. However, after dividing the 46 subjects into two groups based on emphysema or no emphysema, the case number in each group may be small. Perhaps, it is warranted to do this research regarding emphysema after increasing more cases and using or adding another method such as HRCT to diagnose emphysema. I added the notion in the study limitations. |
- The measurement of VD/VT is invasively not convenient. However, the measurement of IC as a marker of DH is easy and convenient. If VD/VT is simply and conveniently evaluated by the method of expired gas analysis only, can the same results be obtained?
|
5. Study limitations. This is possible for normal and patients with CHF but not for COPD because PETCO2 cannot surrogate PaCO2 in these patients. [ref 40, 41] and Liu, Z., et al. Comparison of the end-tidal arterial PCO2 gradient during exercise in normal subjects and in patients with severe COPD. Chest; 107:1218-1224 (1995) |
- The reason why VD/VT has to be measured in addition to the DH is not clear. The worsening of ventilator efficiency may be occurred together with hyperinflation. Is the worsening of VD/VT an independent risk factor for dyspnea and exercise intolerance from DH?
|
4.1. The % of TLC. To obtain VD, VD/VT had to be measured and multiplied by VT. Otherwise, there is no way to obtain VD during exercise. It is not our intention to differentiate whether VD/VT and/or DH together or separately influence exercise intolerance and dyspnea. Instead, we hypothesized that VD and DH volumes are synergic and detrimental to exercise intolerance and dyspnea. VD% alone was poorly related to exercise tolerance and dyspnea. However, adding VD% slightly improved the relationships of DH% and EELV% versus exercise tolerance and dyspnea were. I edited it as recommended. |
- You mentioned that "although the definitions of VD and DH are quite different, this study shows the utility of their combination, and that it may play a role in physiology with regards to the evaluation of exertional dyspnea and exercise capacity in COPD".I have following comments about it:
- EELV and Borg were positively correlated, while VD and Borg were negatively correlated. Please explain why VDEELV which combined those indexes more correlated to dyspnea, even though those indexes have the different correlation to dyspnea respectively.
|
4.1. The % of TLC. VD% and DBorg/DVO2 were insignificantly related but not negatively related, even though the relation was in opposite direction. This opposite direction in relationship does not mean the volume is negative in value. The positive value of VD significantly contributed to EELV and slightly improved the relationship between DBorg/DVO2 and VDEELV. |
- You should also discuss why EELV and VD are negatively correlated. In Figure 2, EELV and VDEELV were expected to be positively correlated because they were almost parallel. Even if the relationship between EELV and VD has not been completely elucidated it in previous study, this is interesting and should be considered.
|
4.1. The % of TLC. VD and EELV were moderately and negatively related. This is because VD and VT were moderately and positively related and VT and EELV were highly and negatively related (r=-0.83, p<0.0001, in our previous report, Scientific Reports | (2019) 9:7514 | https://doi.org/10.1038/s41598-019-43893-1). VD% is positively correlated with VT% that means the larger the VT, the larger the VD. This is because VD is calculated by VD/VT multiplied by VT. For exapmle, in COPD, VD/VT at rest say it is 0.45 and VD/VT at peak exercise say it is 0.35 and VT at rest say it is 500 mL and VT at peak is 1500 mL. It is clear VD at rest is 225 mL and VD at peak exercise is 525 mL. Hence, the larger the VT, the larger the VD, and the smaller the EELV. This is clear that VD is different from EELV or DH but these volumes can be combined and the combinations are slightly more related to exercise capacity and Borg. |
- These results are not discussed enough even though relevant previous studies are cited in the discussion session.
- You mentioned that VD/VT can be estimated from VT. However, the practical relationship between VT and VD/VT is not discussed such as the reason why VD/VT can be estimated from VT.
|
Sorry, I cannot find where I mentioned the notion as above. Please specify the lines where they are. Thanks. |
Minor comments:
- Please add the method of calculation for sample size
|
2.4. Statistical Analysis. Changed as recommended. |
- Table 2 the abbreviations of "EELV" should be deleted because EELV is not included in that table.
|
Table 2. Removed as recommended. |
- The method of evaluation for important outcomes such as DH and Borg should be explained briefly, not only by citing previous own studies
|
2.3.3. and 2.3.4. Changed as recommended. |
- There are four main aim of study in "Introduction" session and four main conclusions in "Discussion" session respectively. You should only mention the points what you would like to present.
|
Follow the suggestion. |
Round 2
Reviewer 3 Report
The authors responded to most of my comments.